

**The Warming Tibetan Plateau improves winter air quality in the Sichuan Basin,**
**China**
Shuyu Zhao[1], Tian Feng[2], Xuexi Tie[1,3*], Zebin Wang[4]
[1]Key Laboratory of Aerosol Chemistry and Physics, SKLLQG, Institute of Earth Environment,
Chinese Academy of Sciences, Xi'an, 710061, China
[2]Department of Geography & Spatial Information Techniques, Ningbo University, Ningbo, 315211,
China
[3]Center for Excellence in Urban Atmospheric Environment, Institute of Urban Environment,
Chinese Academy of Sciences, Xiamen, 361021, China
[4]Northwest Air Traffic Management Bureau, Civil Aviation Administration of China, Xi'an,
712000, China
Corresponding author: tiexx@ieecas.cn



**Key points**

The Tibetan Plateau is rapidly warming, and the temperature has risen by 2 ° C from 2013 to 2017.

The 2 ° C warming of the plateau leads to an increase in PBL height and a decrease in humidity in the Sichuan Basin.

The 2 ° C warming reduces $PM_{2.5}$ concentration in the basin by 25.1 µg m$^{-3}$, of which primary and secondary aerosols are 5.4 µg m$^{-3}$ and 19.7 µg m$^{-3}$, respectively.





**Abstract**
Impacts of global climate change on the occurrence and development of air pollution have attracted
more attentions. This study investigates impacts of the warming Tibetan Plateau on air quality in
the Sichuan Basin. Meteorological observations and ERA-interim reanalysis data reveal that the
Tibetan Plateau has been rapidly warming during the last 40 years (1979-2017), particularly in
winter when the warming rate is approximately twice as much as the annual warming rate. Since
2013, the winter temperature over the plateau has even risen by 2 ° C. Here, we use the WRF-CHEM
model to assess the impact of the 2 ° C warming on air quality in the Sichuan Basin. The model
results show that the 2 ° C warming causes an increase in the Planetary Boundary Layer (PBL)
height and a decrease in the relative humidity (RH) in the basin. The elevated PBL height
strengthens vertical diffusion of $PM_{2.5}$, while the decreased RH significantly reduces secondary
aerosol formation. Overall, $PM_{2.5}$ concentration is reduced by 17.5% (~25.1 μg m$^{-3}$), of which the
reduction in primary and secondary aerosols is 5.4 μg m$^{-3}$ and 19.7 μg m$^{-3}$, respectively. These
results reveal that the recent warming plateau has improved air quality in the basin, to some certain
extent, mitigating the air pollution therein. Nevertheless, climate system is particularly complicated,
and more studies are needed to demonstrate the impact of climate change on air quality in the
downstream regions as the plateau is likely to continue warming.

**Keywords:** climate change, air quality, Tibetan Plateau, WRF-CHEM model



## 1 Introduction

The Tibetan Plateau is known as the third pole because of its high altitude and large area. It is also regarded as an important response region to the Northern Hemisphere, and even global climate due to its sensitivity to climate change. Previous studies on the Tibetan Plateau show that the region was experiencing warming in the second half of the 20th century, especially in the winter months (Kuang and Jiao, 2016; Liu and Chen, 2000; Rangwala et al., 2009). The warming plateau not only plays a significant role in driving the weather and climate change, as well as the ecological system, but also has an important impact on air quality in the downstream regions. Xu et al. (2016) suggest that the thermal anomaly over the Tibetan Plateau obviously increases haze frequency and surface aerosol concentration in central-eastern China.

However, the impacts of climate change on air quality in China are still unclear. Some researches hold the opinion that climate change induced by greenhouse gas emission increases severe haze occurrence and intensity in winter at Beijing, and its impact will continue in the future (Cai et al., 2017; Zou et al., 2017). Similarly, Xu et al. (2017) suggest that climate warming anomaly in the lower and middle troposphere over the continent around the Yangtze River Delta leads to more haze days in winter during recent decades. On the contrary, another opinion suggests that climate change in the past two decades is favorable for air pollution dispersion in northern China via enhancing mid-latitude cold surges in winter (Zhao et al., 2018). If cold surge is strong enough, pollutants would be transported to the downstream regions, causing a better air quality in the upstream region but a worse one in the downstream region. Thus, there may be regional differences in the impact of climate change on air quality.

Previous studies on air pollution in China are concentrated in the developed regions, such as the North China Plain, the Yangtze River Delta and the Pearl River Delta. Few studies have paid attention to the Sichuan Basin, although the region is undergoing severe air pollution, and mean $PM_{2.5}$ concentration is more than 110 μg m$^{-3}$ in winter (Qiao et al., 2019; Tao et al., 2017; Wang et



al., 2018; Yang et al., 2011). Thus, it is necessary to explore the underlying causes that leads to air
pollution in the Sichuan Basin.

The Sichuan Basin locates in the downstream region of the Tibetan Plateau, and its weather
conditions are obviously affected by the plateau (Duan et al., 2012; Hua, 2017; Zhao et al., 2019).
For instance, the foggy weather, southwest vortex and low-level shear line over the basin are closely
associated with the plateau (Zhu et al., 2000). These changes in weather conditions induced by the
plateau undoubtedly affect the development and dispersion of air pollution in the basin, because the
huge terrain can trigger a thermodynamic forcing, which is of great importance for weather
conditions in the surrounding regions (Bei et al., 2016; 2017; Zhao et al., 2015).

This study therefore focuses on how climate change on the Tibetan Plateau affects air quality in the
Sichuan Basin in recent years. Section 3 analyzes the climate change on the Tibetan Plateau in the
past four decades, and especially emphasizes the change in recent five years. In Section 4, we design
two numerical simulations to calculate the impact of climate change on air quality. One is a baseline
simulation, which is constrained by observed surface meteorological parameters and pollutant
concentrations. The other is a sensitivity simulation, which uses the same emission inventory and
meteorological fields as the baseline simulation except for the changed air temperature. We compare
the difference of $PM_{2.5}$ concentrations in these two cases, and also calculate the differences in
meteorological parameters that include winds (wind speed and direction), air temperature, and
relative humidity (RH), as well as the Planetary Boundary Layer (PBL) height. Based on the
differences in $PM_{2.5}$ concentration and meteorological parameters above, we finally explain the
cause-to-effect relationship between climate change on the Tibetan Plateau and the changes in the
PBL height and RH in the Sichuan Basin. Moreover, we calculate the effect of the relationship on
air quality in the Sichuan Basin.

**2 Data and Methods**



**2.1 Observations**

To ensure a robust result, we use two datasets of surface air temperature in this study. One is the
European Center for Medium-Range Weather Forecasts (ECMWF) ERA-Interim monthly mean
reanalysis data (1979-2018), obtained from the website of http://apps.ecmwf.int/datasets/, with the
finest horizontal resolution of 0.125°×0.125°. The other is hourly and monthly mean weather-station
observations from the National Oceanic and Atmospheric Administration (NOAA), which is
available                on                the                website                of
http://gis.ncdc.noaa.gov/map/viewer/#app=clim&cfg=cdo&theme=hourly&layers=1&node=gis.

Figure 1 shows the distribution of weather stations over the Tibetan Plateau, and these weather
stations widely cover the entire plateau. Trends of annual mean and winter surface air temperature
over the plateau are analyzed, and the winter is averaged over 3-month periods (December-January-
February). Additionally, we use ambient air quality data to validate the model performance. Since
2013, the data are released by Ministry of Environmental Protection, China at
http://www.aqistudy.cn/, including hourly $PM_{2.5}$, CO, and $O_3$ mass concentrations. The monitoring
stations for air quality are also shown in Figure 1.

**2.2 Model configuration and experiments**

A state-of-the-art regional dynamical and chemical model (WRF-CHEM model) is used in the
study. The simulation domain covers the Tibetan Plateau and the Sichuan Basin (Figure 1). The
Tibetan Plateau covers about 2.5 million $km^2$, with the averaged elevation of 4500 m, and the
Sichuan Basin covers about 0.16 million $km^2$, with the elevation in the center of the basin less than
1000 m (250 - 700 m). The model is set by a horizontal grid resolution of 9 km (451 × 221 grids),
with 35 vertical sigma levels. The model description in detail is seen by Grell et al. (2005). The
evaluation of the model performance has been conducted by many previous studies (Li et al., 2011a;
Tie et al., 2009; 2007). In this study, we use the Goddard longwave and shortwave radiation



parameterization (Dudhia, 1989), the WSM 6-class graupel microphysics scheme (Hong and Lim,
2006), the Mellor-Yamada-Janji (MYJ) planetary boundary layer scheme (Janjić, 2002), the
unified Noah land-surface model (Chen and Dudhia, 2001) and Monin-Obukhov surface layer
scheme (Janjić, 2002). For chemical schemes, we use a new flexible gas-phase chemical module
and the Community Multiscale Air Quality (CMAQ, version 4.6) aerosol module developed by the
US EPA (Binkowski, 2003). Gas-phase atmospheric reactions of volatile organic compounds
(VOCs) and nitrogen oxide (NOx) use the SAPRC-99 (Statewide Air Pollution Research Center,
version 1999) chemical mechanism. Inorganic aerosols use the ISORROPIA version 1.7, referring
to Li et al. (2011a) and Feng et al. (2016). A $SO_2$ heterogeneous reaction mechanism on aerosol
surfaces involving aerosol water is added (Li et al., 2017a), and $NO_2$ heterogeneous reaction to
produce HONO is also considered (Li et al., 2010). The secondary organic aerosol (SOA)
calculation uses a non-traditional volatility basis-set approach by Li et al. (2011b). The photolysis
rates are calculated by a fast Tropospheric Ultraviolet and Visible (FTUV) radiation transfer model,
in which the impacts of aerosols and clouds on the photochemistry processes are considered (Li et
al., 2011a; Tie et al., 2003; 2005). The wet deposition is calculated by the method used in CMAQ
and the dry deposition follows Wesely (1989).

We use the MIX anthropogenic emission inventory for the year of 2010, and it is available at Multi-
resolution Emission Inventory for China (http://www.meicmodel.org/dataset-mix.html),
consisting of industrial, power, transportation, and agricultural as well as residential sources (Li et
al., 2017b; Zhang et al., 2009). Here, we use a 'top-down' method to constrain the emission
inventory via comparing the simulations with the measurements. The biogenic emissions are online
calculated by the Model of Emissions of Gases and Aerosol from Nature (MEGAN) (Guenther et
al., 2006). Initial and boundary meteorological fields in the model are driven by 6-hour 1° × 1°
NCEP (National Centers for Environmental Prediction) reanalysis data. Chemical lateral
conditions are provided by a global chemistry transport model – MOZART (Model for OZone And
Related chemical Tracer, version 4), with a 6-h output (Emmons et al., 2010; Tie et al., 2005). The
spin-up time of the WRF-CHEM model is 1 day.






Two numerical experiments are performed. One is the baseline simulation in the 2013-2014 winter
(January 2014), and the other is a sensitivity simulation that has an observational increase in air
temperature over the Tibetan Plateau. In other words, the sensitivity simulation uses the same
emission inventory and meteorological conditions as the baseline simulation except that the
temperature fields over the Tibetan Plateau are changed. We set the temperature increment to 2°C
in the sensitivity simulation, because observational temperature increment in winter is about 2°C
from 2013 to 2017 (Table S1). Comparing the difference between the sensitivity simulation and
the baseline simulation, we calculate the impact of the warming Tibetan Plateau on air quality in
the Sichuan Basin.

**3 The warming Tibetan Plateau in the last four decades**

Figure 2 shows the variability and linear trend of surface air temperature at 10 weather stations
over the Tibetan Plateau in winter during the last four decades (1979 - 2017). The winter mean
temperature recorded from all the weather stations exhibits an obvious annual fluctuation and the
linear regression shows a significant rising trend. Clearly, the plateau is continuously undergoing
a warming phase, albeit with regional differences in the warming magnitude. The warming rates
in different regions vary in the range of 0.5 - 1.0°C decade$^{-1}$. Compared with the warming rate of
annual mean temperature (Figure S1), the warming rate in winter is approximately twice as much,
suggesting that the warming in winter is more significant.

Using the ERA-interim reanalysis data, Figure 3 shows the temperature change during the same
period (1979 - 2017). The result is consistent with weather records, showing that air temperature
is significantly rising in most parts of the plateau. The maximal warming rate is around 0.6 - 0.8°C
decade$^{-1}$, appeared in the central and southern plateau. The warming in the rest areas is slighter,
with a rate of 0.3 - 0.6 °C decade$^{-1}$. Particularly, the averaged warming rate in the vast central
plateau reaches about 1.0°C yr$^{-1}$ in recent five years (Figure S2), greater than the warming rate



during the entire 40 years (Figure 3). Both the observation records and reanalysis data evidently
show that the plateau has been warming in the last four decades, and also the warming trend for
recent years is more significant.

From the above temperature change analysis, we notice that there is obviously a positive
temperature anomaly between 2013 and 2017 winters, implying for an accelerating warming over
the plateau. The observational temperature in winter increases by about 2°C between 2013 and
2017. Therefore, the impact of the 2°C warming on air quality in the Sichuan Basin is investigated.
In order to isolatedly assess the effect of a rapid temperature increase and to eliminate the effect of
other factors, a sensitivity study using the WRF-CHEM model is conducted for considering the
2°C temperature increase from the value in 2013 (see Figure 2 and Table S1).

**4 Results and Discussion**

**4.1 Model validation**

To systemically evaluate the model performance on simulation $O_3$, CO and $PM_{2.5}$ mass
concentrations, three statistical indices are used. They are the mean bias (MB), root mean square
error (RMSE), and index of agreement (IOA). The calculation formulas are given in Text S1. The
IOAs of air temperature and RH are 0.85 and 0.79, respectively (Figure S3), suggesting that the
model well captures the diurnal cycle of temperature and the variability of RH. However, the
calculated wind speed is overestimated, especially in the region between the Tibetan Plateau and
the Sichuan Basin. This is because there is a dramatic elevation drop in the region, which makes it
difficult for the model to replicate the observed wind speed and direction.

Figure 4 shows comparisons of hourly $O_3$, CO and $PM_{2.5}$ concentrations between the model
simulations and measurements. The result shows that the simulated CO mean level is close to the
measurement, with a MB of 0.11 mg m$^{-3}$, indicating that the model reasonably reproduces the


meteorological fields and long-range transport. Because the chemical lifetime of CO is relatively
long (~months), the variability of CO is dominantly determined by the meteorological fields and
atmospheric transport process. For the simulation of $O_3$, in addition to the effects of meteorological
fields and atmospheric transport process, its variability is strongly controlled by the photochemical
process. The model result shows that the simulated diurnal cycle of $O_3$ is reasonably agreed with
the measurement, with an IOA of 0.79. There is only a small bias between the simulated and
measured $O_3$ mean concentration. The simulated $O_3$ concentration is 1.7 μg m$^{-3}$ higher than the
measurement, suggesting that both the photochemistry and long-range transport well capture the $O_3$
variability in the region. Finally, the IOA between the simulated and measured $PM_{2.5}$ concentrations
is 0.80, indicating that the aerosol module in the model generally captures the measured $PM_{2.5}$
variation.

However, there are some noticeable discrepancies between the simulations and the measurements.
For instance, the simulated magnitude of $PM_{2.5}$ concentration is larger than the measurement, and
its mean level is underestimated by 13.1 μg m$^{-3}$, less than 10% of the measurement (~153.5 μg m$^{-}$
$^3$). These discrepancies are likely due to the biases in the uncertainties in emission inventory and
small-scale dynamical fields. During the period of Jan 17$^{th}$ to Jan 20$^{th}$, the overestimated $PM_{2.5}$
concentration is mainly caused by the overestimated wind speed and the departure of wind direction
(Figure S3). This is also shown by the overestimation of CO concentration because the observed
northwest wind is not well simulated due to the complicated topography.

**4.2 Change in winter $PM_{2.5}$ concentration over the basin**

To examine impacts of the warming plateau on $PM_{2.5}$ concentration in winter in the Sichuan Basin,
the time series of $PM_{2.5}$ concentrations in the two case simulations (i.e., with and without the 2°C
warming over the plateau) are respectively calculated (Figure 5). The results show that $PM_{2.5}$
concentration in the basin is significantly reduced by an average of 25.1 μg m$^{-3}$ in the case of 2°C
warming. The maximum hourly reduction reaches to 84.6 μg m$^{-3}$ (Figure S4a) and the maximum



percentage reduction is about 64.5% (Figure S4b). Interestingly, the maximum reduction always
occurs while PM$_{2.5}$ concentration reaches a peak value, which suggests that the impact of the
warming plateau is extremely significant during the period of high PM$_{2.5}$ concentration. This result
is similar to previous studies which also point out that extreme weather plays important roles in
affecting air quality (De Sario et al., 2013; Hong et al., 2019; Tsangari et al., 2016; Zhang et al.,
2016). That is to say, the impact of the warming plateau on air quality is apt to be amplified in
extremely high PM$_{2.5}$ concentrations.

To better understand the impact of the warming plateau on PM$_{2.5}$ concentration in the Sichuan
Basin, we also calculate the changes in PM$_{2.5}$ chemical composition in the basin (Figure 6). As a
result, secondary aerosol reduces by 19.7 μg m$^{-3}$, accounting for 78.5% of the total reduction. For
example, the largest reduction is SOA, reducing from 23.2 μg m$^{-3}$ in the base case to 10.8 μg m$^{-3}$
in the warming case. The second reduction is sulfate (31.8 μg m$^{-3}$ in the base case and 28.6 μg m$^{-3}$
in the warming case). The next are nitrate and ammonium (22.3 μg m$^{-3}$ and 19.1 μg m$^{-3}$ in the base
case, and 20.2 μg m$^{-3}$ and 17.5 μg m$^{-3}$ in the warming case).

There are also significant changes in the spatial distribution of PM$_{2.5}$ concentration. Figure 7 shows
the spatial distribution of changes in surface PM$_{2.5}$ concentration and winds after 2°C warming over
the plateau. Apparently, there is a larger decrease in PM$_{2.5}$ concentration in the whole basin, and the
maximum reduction is more than 30 μg m$^{-3}$. By contrast, PM$_{2.5}$ concentration increases by 5 - 15 μg
m$^{-3}$ at the eastern edge of the plateau. Wind patterns show that easterly winds over the basin enhance
while westerly wind over the plateau weaken (Figure S5 and Figure 7). Enhanced easterly winds
cause an increased transport of PM$_{2.5}$ from the basin to the plateau. Moreover, weakened westerly
winds convergent with enhanced easterly winds on the border between the plateau and the basin,
jointly leading to an increase in PM$_{2.5}$ concentration at the eastern edge of the plateau. Additionally,
northerly winds over the basin slightly enhance, conducive to diluting the air and reducing PM$_{2.5}$
concentration. Thus, easterly winds transport and northerly winds dilution are both favorable for a
reduction of PM$_{2.5}$ concentration in the basin. In addition to the wind effect, there are also other





important factors to produce the PM$_{2.5}$ reduction in the basin, such as the PBL height and RH, which
will be analyzed as follows.

**4.3 Impact of PBL height on PM$_{2.5}$ concentration**

Previous studies show that the PBL development plays an important role in diffusing pollutants
(Miao et al., 2017; Su et al., 2018; Tie et al., 2015). Here we calculate the change in the PBL height
due to the 2°C warming over the plateau, and then analyze the effect of the change in PBL height
on PM$_{2.5}$ concentration in the basin.

Our results suggest that the 2°C warming plays different roles in the PBL development over the
plateau and the basin. Due to the 2°C warming, the PBL height decreases in most areas of the plateau,
but rises by 50 - 200 m over the basin (Figure 8). As known, a shallow PBL constrains PM$_{2.5}$ near
the surface via suppressing vertical dispersion (Fan et al., 2011; Iversen, 1984). Conversely, a deep
PBL is favorable for PM$_{2.5}$ diffusion. Thus, we explore the underlying cause that leads to the
difference in the PBL height over the plateau and the basin. Figure 9 shows that vertical profiles of
changes in temperature and winds in the plateau and the basin, because the PBL height is strongly
related to the changes in vertical temperature and wind. The results show that the 2°C warming
causes a maximum warm layer around 1 km above the ground of the plateau. Interestingly, the warm
layer acts as a dome covering 4.5 km above the Sichuan Basin (Figure 9a). Xu et al. (2017) also
finds out a significant warm plume extending from the plateau to the downstream Sichuan Basin
and Yangtze River Delta by use of NCEP/NCAR reanalysis data. This is closely associated with a
sharp topography decrease (from ~ 5 km in the plateau to < 1 km in the basin). In the basin, there is
a decrease in the temperature from the surface to ~ 4 km, with a maximal temperature reduction (1
- 2°C) located at 1.5 km to 3 km above the ground (Figure 9a). As a result, the vertical temperature
gradient increases and the instability also increases in the lower troposphere of the basin, thereby
causing a higher PBL height than that in the non-warming case (Figure 9b). On the contrary, the
change in vertical temperature profile leads to a decreased vertical temperature gradient and



increased thermal stability in the lower troposphere of the plateau, so the PBL height decreases in
the region.

Moreover, the 2°C warming reduces the zonal temperature gradient from the plateau to the basin,
which causes a weakened westerly wind over the plateau. Meanwhile, the east wind strengthens in
the basin. The weakened westerly wind and the strengthened east wind converge on the east side of
the plateau, triggering an ascending motion. The mechanism of the ascending motion is similar to
the plateau "heat pump" effect that implies for a warm updraft forced by a heating plateau (Lau et
al., 2006). Thus, both the ascending motion at the eastern edge of the plateau and the east wind in
the basin strengthen, which is favorable for the development of the PBL. The elevated PBL
facilitates vertical diffusion, leading to a reduction in $PM_{2.5}$ concentration over the basin.
Additionally, the updraft enhances the $PM_{2.5}$ transport from the basin to the plateau, which also
leads to a decrease in $PM_{2.5}$ concentration over the basin.

**4.4 Effect of RH on $PM_{2.5}$ concentration**

In addition to the PBL height, ambient RH is a key factor for secondary aerosol formation (Tie et
al., 2017; Wang et al., 2016). Previous studies indicate that aerosol hygroscopic growth cannot
occurs until the humidity exceeds 50% (Liu et al., 2008). When the humidity is greater than 60%,
hygroscopic growth factor of urban aerosol increases significantly with humidity (Liu et al., 2008).

Figure 10 shows that there is remarkable change in RH in the basin due to the 2°C warming of the
plateau. In the baseline simulation, the RH varies in the range of 40% - 80% over the basin (Figure
10a). However, the RH varies from 40% to 70% in the 2 °C warming simulation (Figure 10b),
suggesting that the basin becomes drier when the plateau is warmer.

The RH comparison between these two numerical simulations reveals that the 2 °C warming causes
a 2.5% - 10% decrease in the RH over the basin (Figure 11). This change in RH has a critical effect
on the secondary aerosol formation. As explained by Tie et al. (2017), the reduction of RH





(especially during the stage of RH from 80% to 70%) causes a significant decrease of hygroscopic
growth on the aerosol surface, resulting in less water surface for producing secondary aerosol, such
as sulfate and nitrate. As a result, the $PM_{2.5}$ concentration decreases in the basin. There are also some
fingerprints of the RH's effect on $PM_{2.5}$ concentration. Firstly, the spatial distributions of RH
reduction and $PM_{2.5}$ concentration reduction have similar patterns (Figure 11 and Figure 7), and the
region with more humidity decrease overlaps the region with more $PM_{2.5}$ decreases. Secondly, as
shown in Figure 6, the changes in $PM_{2.5}$ compositions indicate that the reduced $PM_{2.5}$ concentration
is mainly caused by the decrease in secondary aerosol concentration. Therefore, the RH change
plays an important role for $PM_{2.5}$ concentration in the basin.

**5 Conclusions**

ERA-interim reanalysis data and observation records at 10 weather stations show that the Tibetan
Plateau is significantly warming during the past four decades (1979-2017), particularly in winter.
The temperature increase rate is 0.5°C decade$^{-1}$ to 1.0°C decade$^{-1}$ in winter, approximately twice as
much as the increase rate of annual mean temperature. In recent 5 years (2013-2017), the central
plateau is significantly warming with an increase rate of 1.0°C yr$^{-1}$, encompassing the warming rate
during the entire 40 years. Rapid warming has caused the winter temperature to increase by an
average of 2°C over the entire plateau from 2013 to 2017.

The WRF-Chem model is used to assess the impact of 2°C warming of the plateau on air quality
over the downstream Sichuan Basin. The most significant impact of the 2°C warming on $PM_{2.5}$
concentration in the basin is via reducing relative humidity and increasing PBL height. A lower
ambient humidity decreases aerosol hygroscopic growth, which weakens secondary aerosol
formation and leads to a significant reduction in secondary aerosol concentration. Moreover, the
2°C warming induces an increase in vertical temperature gradient over the basin, strengthening
turbulence mixing and elevating PBL height. The elevated PBL height is favorable for vertical
diffusion that causes a reduction of $PM_{2.5}$ in the basin. Additionally, the uplift effect by an enhanced



ascending motion at the eastern edge of the plateau also contributes to $PM_{2.5}$ reduction within the
basin.

In summary, the 2°C warming over the plateau in recent five years comprehensively induces a rising
PBL height and a drying ambient air over the basin, which greatly reduces $PM_{2.5}$ secondary
compositions. On average, $PM_{2.5}$ concentration reduces by 25.1 μg m$^{-3}$, of which the primary and
secondary aerosols decrease by 5.4 μg m$^{-3}$ and 19.7 μg m$^{-3}$, respectively. Since the plateau is likely
to continue warming, in-depth understanding to climate change on the Tibetan Plateau is required.
Long-term $PM_{2.5}$ monitoring is also needed to validate the impact of the warming plateau on air
quality.

*Data availability*. The data used in this study are available from the corresponding author upon
request (tiexx@ieecas.cn).
*Supplement*. Supplemental materials to this article can be found online at http://xxxxxx
*Author contributions*. XX designed research, and revised the final paper. SY performed research,
and wrote the paper. XX and SY provided financial support. TF validated the model, modified the
chart code and reviewed the paper. ZB collected and analyzed the weather-stations data.
*Competing interests*. The authors declare that they have no conflict of interest.
*Acknowledgements*. This work is supported by the National Natural Science Foundation of China
(Nos. 41430424, 41730108 and 41807307) and the West Light Foundation of the Chinese Academy
of Sciences (Nos. XAB2016B04). We also would like to acknowledge European Center for
Medium-Range Weather Forecasts (ERA-interim) for reanalysis data which are freely obtained by
a following registration on the website http://apps.ecmwf.int/datasets/. Ambient weather-station
observations are obtained from the National Oceanic and Atmospheric Administration (NOAA),
http://gis.ncdc.noaa.gov/map/viewer/#app=clim&cfg=cdo&theme=hourly&layers=1&node=gis.
The hourly ambient surface $O_3$, CO and $PM_{2.5}$ mass concentrations are real-timely released by
Ministry of Environmental Protection, China on the website http://www.aqistudy.cn/, freely
downloaded from http://106.37.208.233:20035/. The MEIC-2012 (Multi-resolution Emission
Inventory for China) anthropogenic emission inventory is available on the website,
http://www.meicmodel.org. The authors also thank anonymous reviewers for their helpful
comments and suggestions.



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



# Figure captions

**Figure 1** (a) Location map of the Tibetan Plateau (the region surrounded by the dark line) and the Sichuan Basin (the region surrounded by the gray line). (b) The model domain and the distribution of weather stations marked in the triangles over the Tibetan Plateau and air quality stations marked in the circles over the Sichuan Basin.

**Figure 2** Trends of observational winter (Dec-Jan-Feb) mean temperature anomaly recorded by 10 weather stations over the Tibetan Plateau during the last four decades (1979-2017).

**Figure 3** Trends of ERA-interim reanalysis winter mean temperature over the Tibetan Plateau from 1979 to 2017. The dotted regions show statistical significance with 95% confidence level ($p$-value $< 0.05$) from the Student's $t$ test.

**Figure 4** Comparison between the observed (black dots) and simulated (blue line) hourly $O_3$ ($\mu g\ m^{-3}$), CO ($mg\ m^{-3}$) and $PM_{2.5}$ mass concentrations ($\mu g\ m^{-3}$) over the Sichuan Basin in January 2014.

**Figure 5** Time series of $PM_{2.5}$ concentrations over the Sichuan Basin, the baseline simulation is selected in January 2014 and the sensitivity simulation in which 2°C warming occurs over the Tibetan Plateau relative to the baseline simulation.

**Figure 6** Comparison of chemical composition of $PM_{2.5}$ mass concentrations between the (a) baseline simulation and (b) sensitivity simulation over the Sichuan Basin. The yellow fan-shaped area presents the component of secondary aerosol, and the rest presents primary aerosol.

**Figure 7** Difference in spatial distributions of surface $PM_{2.5}$ concentrations and winds between the sensitivity simulation and baseline simulation. The negative shows $PM_{2.5}$ concentrations decrease when the Tibet is 2°C warming, and the positive shows $PM_{2.5}$ concentrations increase when the Tibet is 2°C warming.

**Figure 8** Spatial change in the PBL height induced by 2°C warming over the Tibet. The positive shows the PBL height increases while the negative shows the PBL height decreases.

**Figure 9** Vertical profiles of changes in temperature (color shading and gray contour) and winds (arrows) along 30°N in January 2014. The gray shaded area presents topography. The green box for the Sichuan Basin, and the red solid (baseline simulation) and dash (sensitivity simulation) lines for the PBL height. (a) The Tibet and Sichuan Basin, and (b) The Sichuan Basin.

**Figure 10** Comparison of spatial distributions of relative humidity (RH) between the (a) baseline simulation and (b) sensitivity simulation over the Tibet and Sichuan Basin.

**Figure 11** Spatial change in the relative humidity after the Tibet becomes 2°C warming. The positive shows the RH increases while the negative shows the RH decreases.



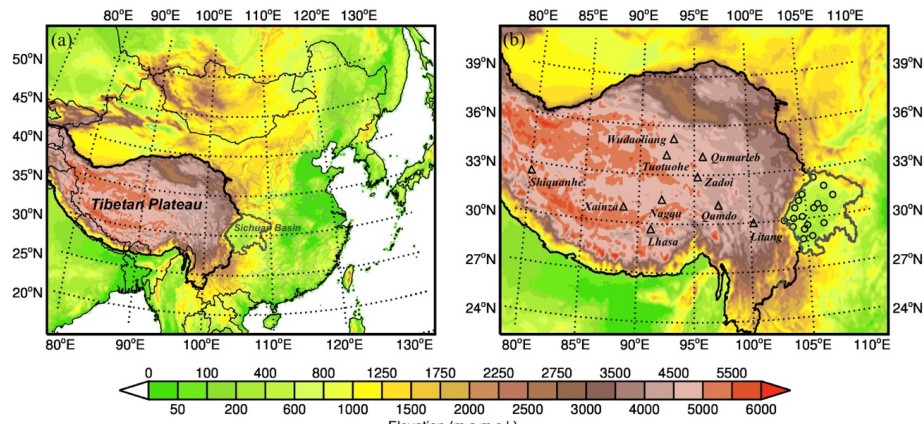

**Figure 1** (a) Location map of the Tibetan Plateau (the region surrounded by the dark line) and the Sichuan Basin (the region surrounded by the gray line). (b) The model domain and the distribution of weather stations marked in the triangles over the Tibetan Plateau and air quality stations marked in the circles over the Sichuan Basin.

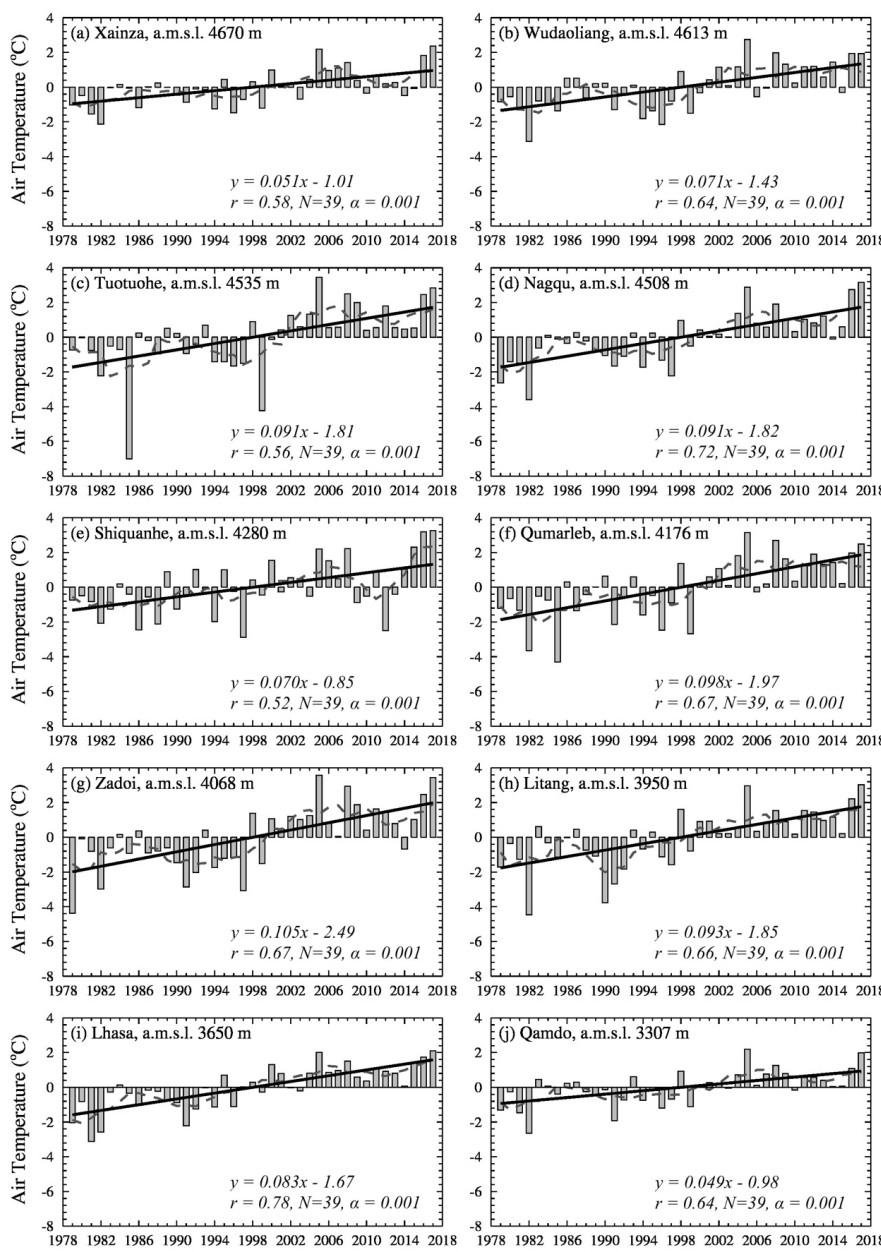

**Figure 2** Trends of observational winter (Dec-Jan-Feb) mean temperature anomaly recorded by 10 weather stations over the Tibetan Plateau during the last four decades (1979-2017).

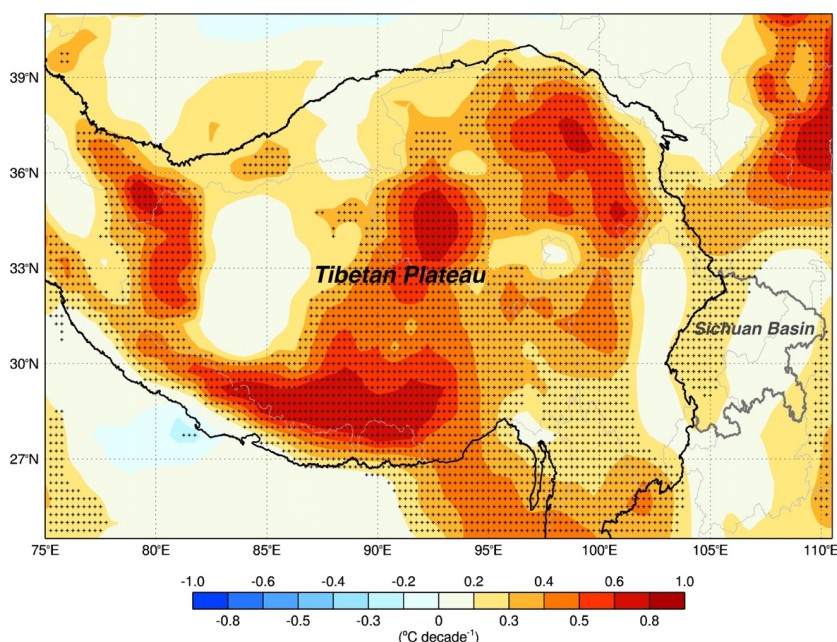

**Figure 3** Trends of ERA-interim reanalysis winter mean temperature over the Tibetan Plateau from 1979 to 2017. The dotted regions show statistical significance with 95% confidence level ($p$-value $< 0.05$) from the Student's $t$ test.

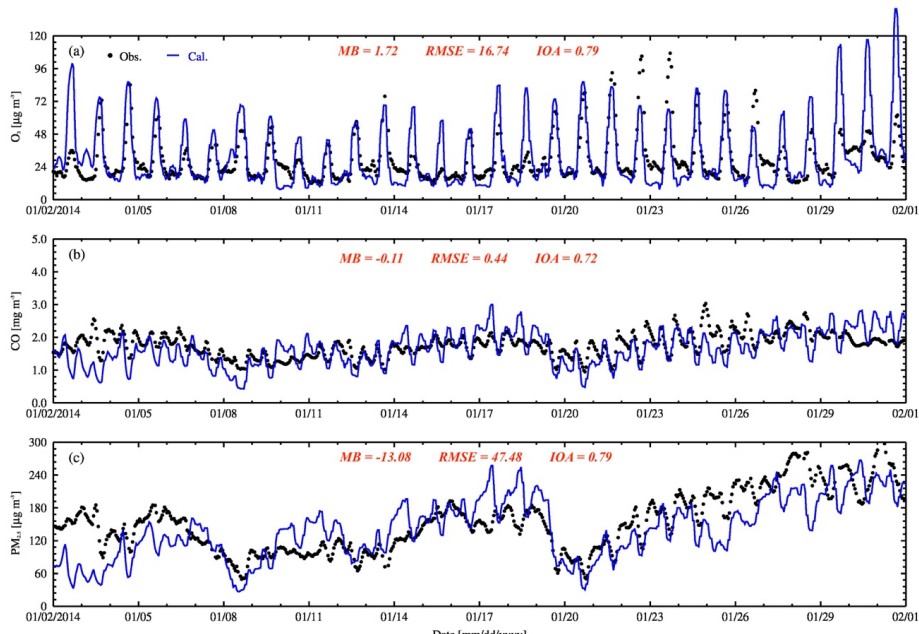

**Figure 4** Comparison between the observed (black dots) and simulated (blue line) hourly $O_3$ ($\mu g\ m^{-3}$), CO ($mg\ m^{-3}$) and $PM_{2.5}$ mass concentrations ($\mu g\ m^{-3}$) over the Sichuan Basin in January 2014.





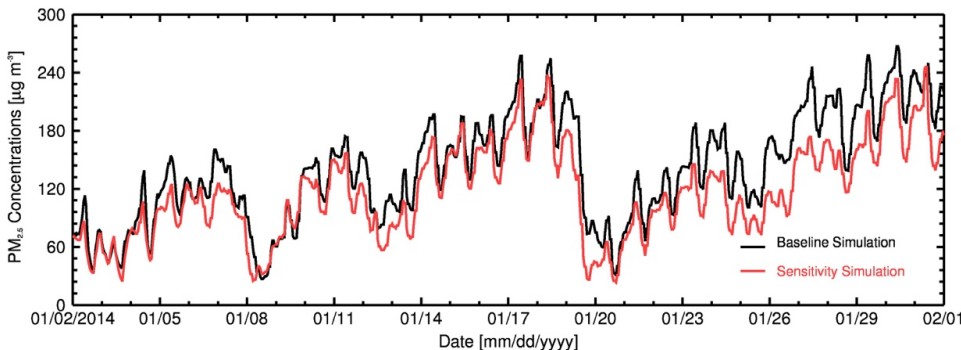

**Figure 5** Time series of PM$_{2.5}$ concentrations over the Sichuan Basin, the baseline simulation is selected in January 2014 and the sensitivity simulation in which 2°C warming occurs over the Tibetan Plateau relative to the baseline simulation.



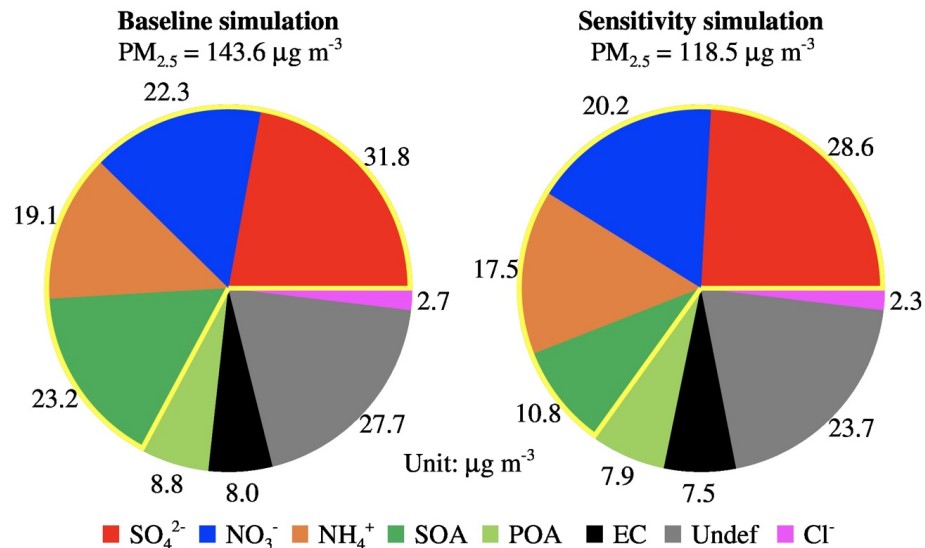

**Figure 6** Comparison of chemical composition of PM$_{2.5}$ mass concentrations between the (a) baseline simulation and (b) sensitivity simulation over the Sichuan Basin. The yellow fan-shaped area presents the component of secondary aerosol, and the rest presents primary aerosol.

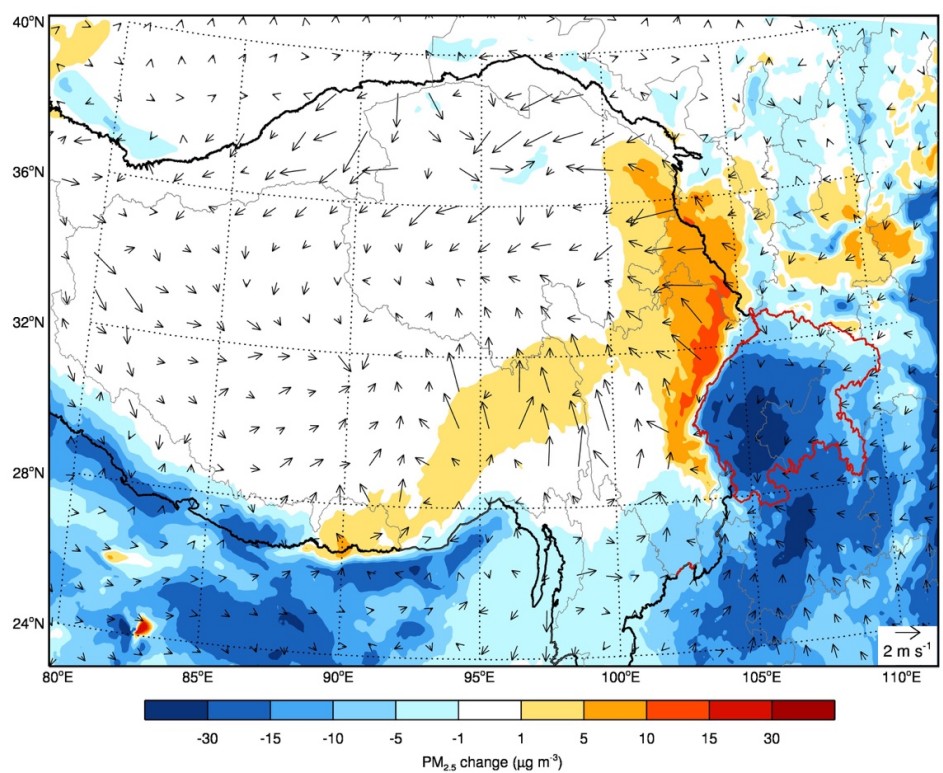

**Figure 7** Difference in spatial distributions of surface PM$_{2.5}$ concentrations and winds between the sensitivity simulation and baseline simulation. The negative shows PM$_{2.5}$ concentrations decrease when the Tibet is 2°C warming, and the positive shows PM$_{2.5}$ concentrations increase when the Tibet is 2°C warming.

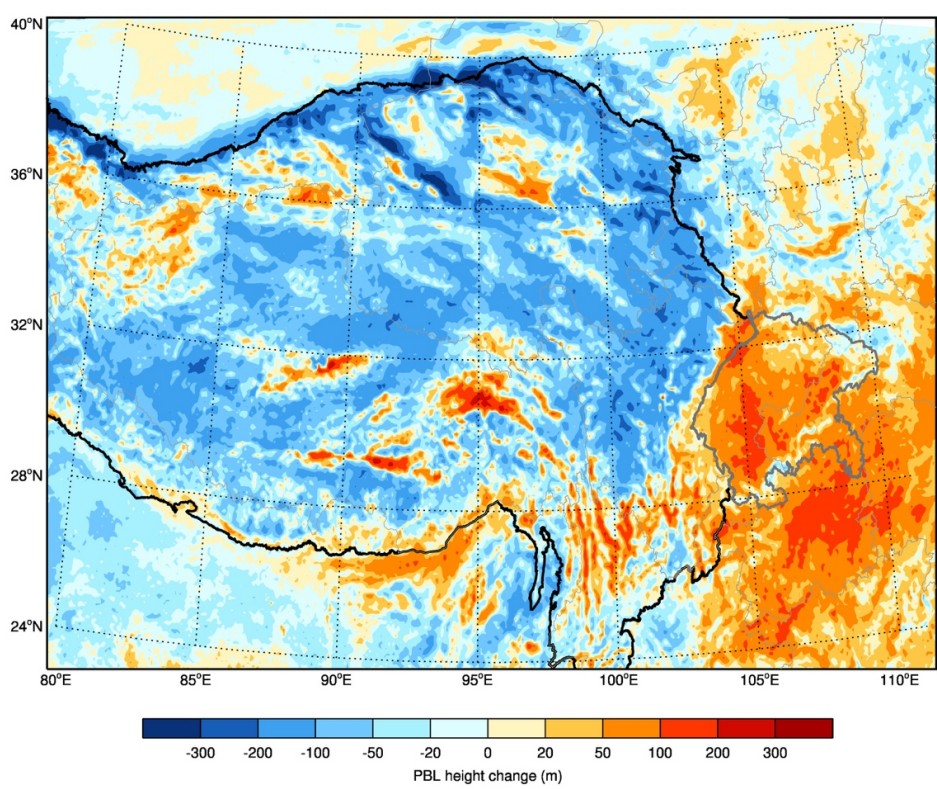

**Figure 8** Spatial change in the PBL height induced by 2°C warming over the Tibet. The positive shows the
PBL height increases while the negative shows the PBL height decreases.



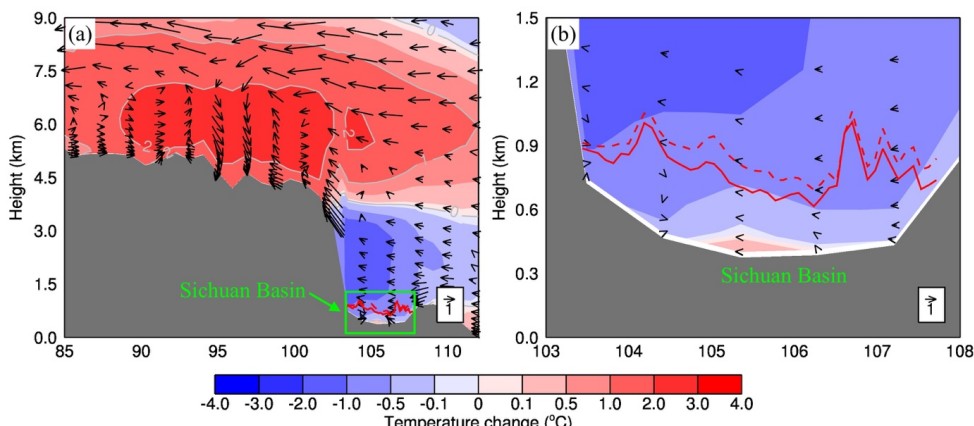

**Figure 9** Vertical profiles of changes in temperature (color shading and gray contour) and winds (arrows) along 30°N in January 2014. The gray shaded area presents topography. The green box for the Sichuan Basin, and the red solid (baseline simulation) and dash (sensitivity simulation) lines for the PBL height. (a) The Tibet and Sichuan Basin, and (b) The Sichuan Basin.





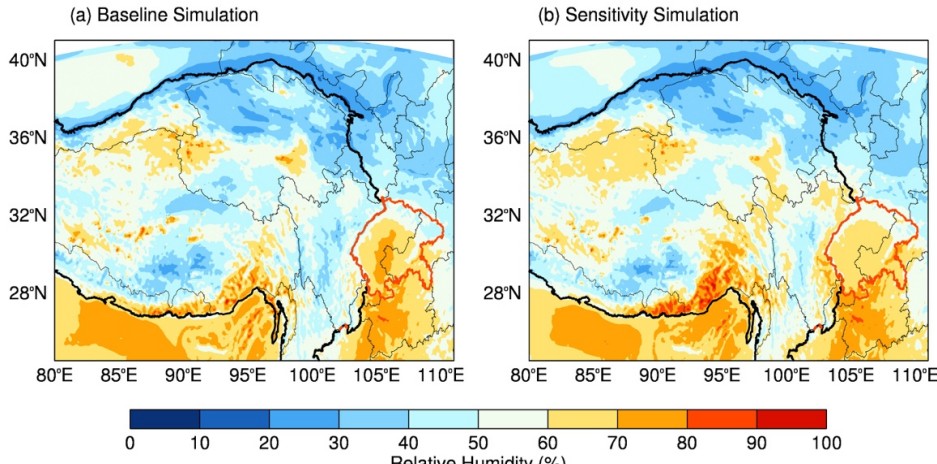

**Figure 10** Comparison of spatial distributions of relative humidity (RH) between the (a) baseline simulation and (b) sensitivity simulation over the Tibet and Sichuan Basin.





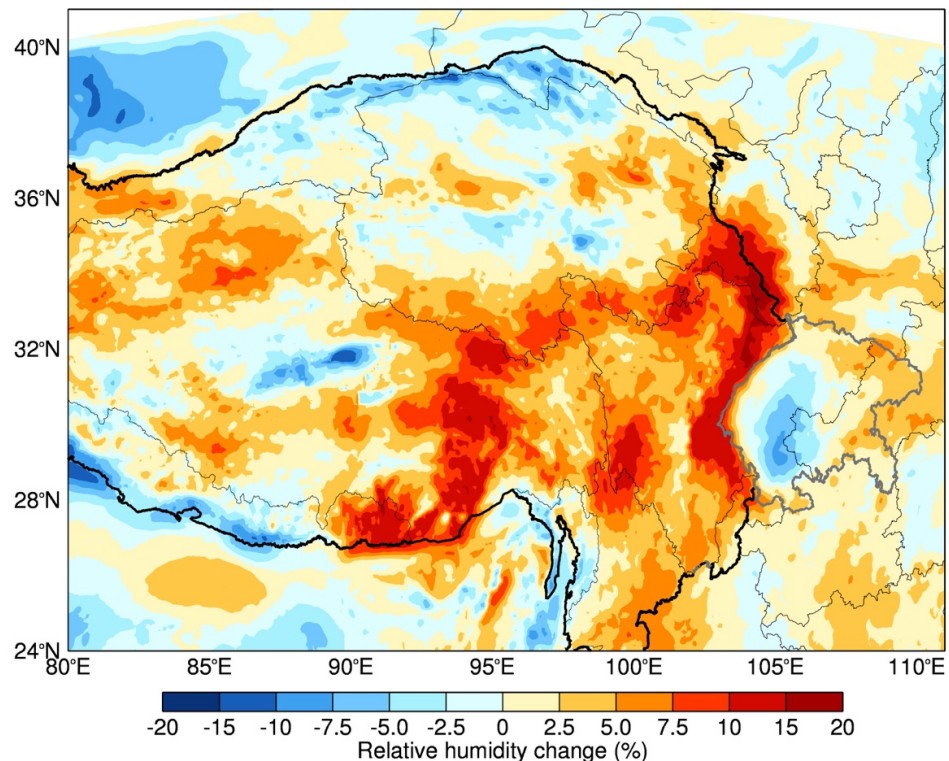

3    **Figure 11** Spatial change in the relative humidity after the Tibet becomes 2°C warming. The

4    positive shows the RH increases while the negative shows the RH decreases.

