# Peer review of "The Warming Tibetan Plateau improves winter air quality in the Sichuan Basin, 1 2 China 3 Shuyu Zhao1, Tian Feng2, Xuexi Tie1,3\*, Zebin Wang4 4 5 6 1Key Laboratory of Aerosol Chemistry and Physics, SKLLQG, Institute of Earth Environment, 7 Chin"

_Atmospheric Chemistry and Physics, 2020_

## Referee Comment (RC1) · Anonymous Referee #1 · 5 May 2020

General comments:

In this manuscript, the authors focus on the effect of warming Tibetan Plateau on air quality in the Sichuan Basin, China. Specifically, they address the 2 ° C warming causes an increase in the PBL height and a decrease in the relative humidity in the basin. The elevated PBL height strengthens vertical diffusion of PM2.5, while the decreased RH significantly reduces secondary aerosol formation. The authors highlight that the recent warming plateau has improved air quality in the basin. The results of this work are based on the WRF-Chem simulations and extensive observation. The analysis is mostly sound, the manuscript is well written, but some details need clarify. I recommend a minor revision with my comments listed below.

Specific comments: 1. In line151, please further explain what does "'top-down'

method" means here and how to use the 'top-down' method to constrain the emission inventory via comparing the simulations with the measurements?

2. In line 164-165, in the configuration of the sensitivity simulation, how to set the temperature increment to 2K? Is it just increase the temperature in all levels and all grids of the model above Tibetan Plateau (TP)? Does the 2K increment set at the beginning of model simulation or need nudging in every step of the simulation? Are the temperature increment same in verticals or just at the surface?

3. In line 231-232, is it correct here "the overestimated PM2.5 concentration is mainly caused by the overestimated wind speed"? Or underestimated wind speed?

4. Could you further explain the thermodynamic reasons of the winds and PBLH changes due to 2K warming over TP in figure 7 and the description in line 263-269 "easterly winds over the basin enhance while westerly wind over the plateau weaken……northerly winds over the basin slightly enhance,"?

5. In line 293-295, similarly, could you further explain the mechanism of "a maximal temperature reduction located at 1.5 km to 3 km above the ground (Figure 9a)"?

6. Related to comments 4 and 5, the paragraph from line 302-311 did not make very clear discussion on the changes of wind and temperature gradient. I suggest the comparison of the changes of pressure-difference between TP and basin, and see the circulation changes could easily explain the issues in comments 4 and 5.

7. I don't think the ascending motion in this study is similar to the plateau "heat pump" effect raised by Lau (2016).

Technical corrections:

1. I am misleading by the figure 6 in the first look and regards they are pie charts in percentage of species. Plot them as columns could be better.

2. Setting figure 11 as figure 10c is reasonable.

---

## Referee Comment (RC2) · Anonymous Referee #3 · 18 Jun 2020

General Comments
 This paper investigated the role of warming Tibetan Plateau on winter air quality in the Sichuan Basin, China. This paper has indicated that the air temperature in winter over the TP has risen by 2 degrees from 2014 to 2017. Then the authors used sensitivity experiments to examine the influence of the waring TP on air quality in the Sichuan Basin. This paper is well written and well organized. However, this manuscript has not provided any physical explanations for the linkage between warming TP and less air quality in Sichuan Basin. In fact, I doubt that the relation between warming TP and less air pollution is not a cause-and-effect relation other than a companion relation caused by atmospheric circulation. Based on the following comments, I will not recommend publication for this manuscript at current situation. Of course, the resubmission is encouraged.

[Figure]

Major comments: 1, The description on the experiment design is too simple to be understood. How the authors set the temperature increment to 2 degrees ? Only stations over the TP or all grids in the domain of the TP ? please clarify this issue. 2, Please clarify the mechanism that the warming TP causes less air pollution in the Sichuan Basin. Please make sure whether the warming TP influence large-scale atmospheric circulation through air-land interaction? I think that the warming TP is a result other than a cause. 3, significance testing is important for your results. Please make some significance test for your results. For example, Fig. 5 and Fig. 6 show the difference between observations and simulations. Whether the difference between them is significant ? Minor comments: 1, Fig. 7, please indicate the information of winds.

---

## Author Comment (AC1) · 1 Jul 2020

**Reply to Anonymous Referee #3**

Thanks for the reviewer's helpful comments. We have given our point-to-point response to your comments in the revised manuscript.

**General comments:**

This paper investigated the role of warming Tibetan Plateau on winter air quality in the Sichuan Basin, China. This paper has indicated that the air temperature in winter over the TP has risen by 2 degrees from 2014 to 2017. Then the authors used sensitivity experiments to examine the influence of the waring TP on air quality in the Sichuan Basin. This paper is well written and well organized. However, this manuscript has not provided any physical explanations for the linkage between warming TP and less air quality in Sichuan Basin. In fact, I doubt that the relation between warming TP and less air pollution is not a cause-and-effect relation other than a companion relation caused by atmospheric circulation. Based on the following comments, I will not recommend publication for this manuscript at current situation. Of course, the resubmission is encouraged.

**Major comments:**

**Comment 1.** The description on the experiment design is too simple to be understood. How the authors set the temperature increment to 2 degrees? Only stations over the TP or all grids in the domain of the TP? please clarify this issue.

**Response:** We have added the detailed description of the 2°C increment settings. In the sensitivity simulation, we set the 2°C increment at all grids in the domain of the TP. The text is "*According to the meteorological records at weather stations, surface air temperature risen by an average of 2°C from 2013 to 2017 over the Tibetan Plateau (Table S1). ERA-interim reanalysis data also show that the troposphere (600hPa - 250hPa) over the plateau is warming during the 2013-2017 period, and the temperature increment shows a parabolic pattern with the altitude, by an average increase of ~2°C (Figure S1). Thus, we design a sensitivity simulation, with a temperature increase of 2°C in the troposphere over the plateau. In the model, we set to the 2°C warming at all grids covering the plateau (the region surrounded by the dark line in Figure 1b) in the initial and boundary fields. In order to ensure a persistent influence of the 2°C warming, we drive the initial field with a 2°C increment every day. Then, by comparing the difference between the sensitivity simulation and the baseline simulation, we determine the impact of the 2°C warming over the Tibetan Plateau on air quality in the Sichuan Basin.*" In Lines 167 - 178.

[Figure]

**Figure S1** Vertical profile of temperature change along the longitude (80°E -100°E) covering the plateau at 28°N, 30°N, 32°N and 34°N, ΔT is calculated by the annual temperature increase rate from 2013 to 2017 multiplying by the number of years (*N* = 5). Noted that the temperature in the troposphere over the Tibetan Plateau (600 hPa - 250 hPa) is inhomogeneously warming by 0 - 4 °C from 2013 to 2017, and we take an average warming increase of 2 °C.

**Comment 2.** Please clarify the mechanism that the warming TP causes less air pollution in the Sichuan Basin. Please make sure whether the warming TP influence large-scale atmospheric circulation through air-land interaction? I think that the warming TP is a result other than a cause.

**Response:** We have added the analysis of pressure gradient to explain the mechanism between the warming TP and less air pollution in the Sichuan Basin (Figure 8, a new figure). In our study, we focus on how the warming TP affects air pollution via changing winds, temperature and the PBL height as well as RH in the Sichuan Basin.

The text is as follows: "*We further compare the difference in the surface pressure between the baseline and sensitivity simulations, and find out that surface pressure over the plateau and the basin all decreases when the plateau warms by 2°C (Figure 8a and 8b). Over the plateau, the pressure drop has a decrease characteristic from west to east (Figure 8c), which results in a decreased pressure gradient and a weakened westerly wind. While in the basin, the pressure drop is less than the plateau. This leads to an increased pressure gradient from the basin to*

*the plateau, inducing an intensified easterly wind. The enhanced easterly wind causes an increased transport of $PM_{2.5}$ from the basin to the plateau. On the other hand, the weakened westerly wind and the enhanced easterly wind are convergent at the border between the plateau and the basin (Figure 7), jointly leading to an increase in $PM_{2.5}$ concentration at the eastern edge of the plateau."* in Lines 279 - 289.

*"After the 2°C warming, surface pressure decreases in the basin (Figure 8), which produces more convergent airflow (as shown in Figure 7). The strengthened convergent airflow induces an intensified ascending motion, conducive to a reduction of temperature in the basin. As a result, the zone where the maximal temperature drop appears, overlaps with the zone with the maximal ascending motion. Furthermore, the intensified updraft increases the vertical temperature gradient and the instability in the lower troposphere of the basin, thereby causing a higher PBL height than that in the non-warming case (Figure 10b). On the contrary, the change in vertical temperature profile leads to a decreased vertical temperature gradient and increased thermal stability in the lower troposphere of the plateau, in which the PBL height decreases.*

*On the other hand, the convergent airflows by a weakened westerly wind over the plateau and a strengthened easterly wind in the basin (shown in Figure 8) triggers an ascending motion on the east side of the plateau, which is also beneficial to the development of the PBL height in the basin. Consequently, the elevated PBL facilitates vertical diffusion, leading to a reduction in $PM_{2.5}$ concentration over the basin."* in Lines 318 - 332.

[Figure]

**Figure 8** Comparison of spatial distributions of sea level pressure (SLP) between the (a) baseline simulation and (b) sensitivity simulation over the Tibetan Plateau and Sichuan Basin. (c) The SLPs over the plateau and basin decrease while the plateau becomes 2°C warming.

**Comment 3.** significance testing is important for your results. Please make some significance test for your results. For example, Fig. 5 and Fig. 6 show the difference between observations and simulations. Whether the difference between them is significant?

**Response:** Thanks for your suggestions, and we have added the Student's *t test* to validate the significant difference between observations and simulations in the revised manuscript. Results show that the difference is extremely significant, and here we have added the *p*-value ($p <$ 0.001) in Figure 5, rather than the exact value, because the *p*-value ($p = 5.76E-19$) is far less than 0.001. The related text is "*The results show that PM$_{2.5}$ concentration in the basin is significantly reduced by an average of 25.1 μg m$^{-3}$ in the case of 2°C warming, with a confidence level of 99.9% (p < 0.001).*"

Figure 6 calculates monthly-averaged concentrations of chemical composition in PM$_{2.5}$. To

calculate the significance of every chemical composition, we use raw data in Figure 5, because $PM_{2.5}$ concentration is the sum of concentrations of $SO_4^{2-}$, $NO_3^-$, $NH_4^+$, $Cl^-$, SOA, POA, EC and Undef in Figure 6. Results show that differences in most of the chemical composition are extremely significant ($p < 0.001$) except that the EC is more significant ($0.001 < p = 0.0011 < 0.01$). The $p$-values of every chemical composition are summarized as Table S2 in Supplemental materials, not shown in Figure 6. The Table S2 and its related text is below: "*Significance testing of the difference in every chemical composition between the baseline and sensitivity simulations are also given in Table S2. The p-values of most chemical composition in $PM_{2.5}$ are far less than 0.001 except that the p-value of EC is 0.0011 (Table S2), implying for an extremely significant reduction of every chemical composition in $PM_{2.5}$ within the basin when the plateau warms by 2°C.*"

*Table S2 Significance differences in concentrations of chemical composition in $PM_{2.5}$ between the baseline simulation and sensitivity simulation. The p-value of every chemical composition is followed.*

| Chemical composition | $SO_4^{2-}$ | $NO_3^-$ | $NH_4^+$ | $Cl^-$ | SOA | POA | EC | Undef |
|---|---|---|---|---|---|---|---|---|
| $p$-value | 2.78E-04 | 5.05E-06 | 6.84E-05 | 3.29E-04 | 6E-130 | 2.14E-06 | 0.0011 | 2.63E-15 |

**Minor comments:**

**Comment 4:** Fig. 7, please indicate the information of winds.

**Response:** Added. *"On the other hand, the weakened westerly wind and the enhanced easterly wind are convergent at the border between the plateau and the basin (Figure 7), jointly leading to an increase in $PM_{2.5}$ concentration at the eastern edge of the plateau. Additionally, northerly winds over the basin slightly enhance, conducive to diluting the air and reducing $PM_{2.5}$ concentration."* In Lines 286 - 290.

---

## Author Comment (AC2) · 1 Jul 2020

June 20, 2020

Dear Editor,

Thanks for your efforts on this manuscript. We have received comments from the reviewers of our manuscript, and we would like to thank the reviewers for their careful reading and their insightful comments. To address the reviewers' comments, we have revised the manuscript, and the revised text is highlighted in red.

Best Regards,
Xuexi Tie

**Reply to Anonymous Referee #1**

Thanks for the reviewer's helpful comments. We have given our point-to-point response to your comments and suggestions in the revised manuscript. To carefully address the comments of the reviewer, we add more content and figures. We would like to think that the revised manuscript is greatly improved after addressing the reviewer's comments.

**General comments:**

In this manuscript, the authors focus on the effect of warming Tibetan Plateau on air quality in the Sichuan Basin, China. Specifically, they address the 2℃ warming causes an increase in the PBL height and a decrease in the relative humidity in the basin. The elevated PBL height strengthens vertical diffusion of PM2.5, while the decreased RH significantly reduces secondary aerosol formation. The authors highlight that the recent warming plateau has improved air quality in the basin. The results of this work are based on the WRF-Chem simulations and extensive observation. The analysis is mostly sound, the manuscript is well written, but some details need clarify. I recommend a minor revision with my comments listed below.

**Specific comments:**

**Comment 1.** In line151, please further explain what does "'top-down' method" means here and how to use the 'top-down' method to constrain the emission inventory via comparing the simulations with the measurements?

**Response:** The 'top-down' method is to compare the simulated value with the observed value time and again until the simulated values, including the averaged level and the trend, are close to the observed ones. Generally, we use mean bias (MB), root mean square error (RMSE), and index of agreement (IOA) to evaluate the model performance. The higher the IOA, the closer the simulated value is to the observed value. In this study, the statistical indices of agreement (IOAs) of pollutants ($O_3$, CO and $PM_{2.5}$) are greater than 0.7.

The 'bottom-up' emission inventory used in this study is constructed by national and provincial emission factors and activity data based on a statistical approach, so it is difficult to obtain accurate activity data. Also, the emission factors representative at a local level is difficult to

measured. Therefore, the spatial pattern of the inventory at a local level needs to be improved. In addition, the 'bottom-up' emission inventory is not updated every year. In practice, the 'bottom-up' emission inventory is used to drive the model, and the 'top-down' method is used to constrain the emission. Top-down constraints on emissions is helpful to improve the accuracy of the 'bottom-up' emission inventory. The detailed introduction of these two approaches are referred to Zhang et al. (2009) and Fu et al. (2012).

In the revised version, we have added a brief introduction to the 'top-down' method, and the text is "*The emission inventory is constructed by a 'bottom-up' approach based on national and provincial activity data and emission factors. To improve the emission inventory accuracy, we use a 'top-down' method here to constrain the emission inventory. We compare the simulated value with the measured value time and again until the simulations are close to the measurements.*" In lines 151 - 155.

Fu, T. M., Cao, J. J., Zhang, X. Y., Lee, S. C., Zhang, Q., Han, Y. M., et al. (2012). Carbonaceous aerosols in China: top-down constraints on primary sources and estimation of secondary contribution. *Atmospheric Chemistry and Physics*, *12*(5), 2725–2746. http://doi.org/10.5194/acp-12-2725-2012

Zhang, Q., Streets, D. G., Carmichael, G. R., He, K. B., Huo, H., Kannari, A., et al. (2009). Asian emissions in 2006 for the NASA INTEX-B mission. *Atmospheric Chemistry and Physics*, *9*(14), 5131–5153. http://doi.org/10.5194/acp-9-5131-2009

**Comment 2.** In line 164-165, in the configuration of the sensitivity simulation, how to set the temperature increment to 2K? Is it just increase the temperature in all levels and all grids of the model above Tibetan Plateau (TP)? Does the 2K increment set at the beginning of model simulation or need nudging in every step of the simulation? Are the temperature increment same in verticals or just at the surface?

**Response:** We have given a detailed description for the 2K sensitivity simulation in the revised version. According to the ERA-interim reanalysis data, the warming is only happening in the troposphere (600 hPa - 250 hPa). As a result, in the sensitivity simulation, we set the 2K increment in the troposphere (600 hPa - 250 hPa) over the Tibetan Plateau. In order to ensure a persistent influence of the 2K increment, we add the 2K increment at the initial and boundary conditions of the model, and also drive the initial condition with a 2K increment every day.

These texts are added in the revised manuscript. "*According to the meteorological records at weather stations, surface air temperature risen by an average of 2°C from 2013 to 2017 over the Tibetan Plateau (Table S1). ERA-interim reanalysis data also show that the troposphere (600hPa - 250hPa) over the plateau is warming during the 2013-2017 period, and the temperature increment shows a parabolic pattern with the altitude, by an average increase of ~2°C (Figure S1). Thus, we design a sensitivity simulation, with a temperature increase of 2°C in the troposphere over the plateau. In the model, we set to the 2°C warming in the initial and boundary fields. In order to ensure a persistent influence of the 2°C warming, we drive the initial field with a 2°C increment every day. Then, by comparing the difference between the sensitivity simulation and the baseline simulation, we determine the impact of the 2°C warming over the Tibetan Plateau on air quality in the Sichuan Basin.*" In Lines 167 - 178.

[Figure]

**Figure S1** Vertical profile of temperature change along the longitude (80°E -100°E) covering the plateau at 28°N, 30°N, 32°N and 34°N, ΔT is calculated by the annual temperature increase rate from 2013 to 2017 multiplying by the number of years ($N = 5$). Noted that the temperature in the troposphere over the Tibetan Plateau (600 hPa - 250 hPa) is inhomogeneously warming by 0 - 4 °C from 2013 to 2017, and we take an average warming increase of 2 °C.

**Comment 3.** In line 231-232, is it correct here "the overestimated PM2.5 concentration is mainly caused by the overestimated wind speed"? Or underestimated wind speed?

**Response:** Yes. We have explained that the overestimated PM$_{2.5}$ concentration here is mainly related to the wind departure in detail, including of an overestimated wind speed and a departure of wind direction. Figure S4 shows that the simulated temperature and humidity are well consistent with the observed, but the simulated winds are not consistent with the observed. Observational wind speed concentrates in the range of 1 - 2 m s$^{-1}$ (the average wind speed is 1.3 m s$^{-1}$), obviously lower than the simulated wind speed (mostly higher than 2 m s$^{-1}$, the average wind speed is 2.0 m s$^{-1}$,)). The observed prevailing wind is northerly wind while the simulated is mainly easterly wind. Figure S6a shows that PM$_{2.5}$ concentration is lower in the north to Sichuan Basin while higher in the east to the basin. Therefore, the simulated high PM$_{2.5}$ concentration is mainly caused by a wind departure, which results in a false transport from the east to the basin. To clarify the explanation, we have revised the text as follows "*During the period of Jan 17$^{th}$ to Jan 20$^{th}$, the observed wind speed concentrates in the range of 1 - 2 m s$^{-1}$, with an average of 1.3 m s$^{-1}$, while the simulated wind speed is obviously higher, with an average of 2.0 m s$^{-1}$ (Figure S3). The observed prevailing wind is northerly wind while the simulated prevails easterly wind. Figure S6a shows that PM$_{2.5}$ concentration is lower in the north to the Sichuan Basin while higher to in the east to the basin. Therefore, the overestimated PM$_{2.5}$ concentration is mainly caused by the departure of winds, which results in a false transport from the east to the basin.*" In lines 241 - 249.

**Comment 4.** Could you further explain the thermodynamic reasons of the winds and PBLH changes due to 2K warming over TP in figure 7 and the description in line 263- 269 "easterly winds over the basin enhance while westerly wind over the plateau weaken... . ..northerly winds over the basin slightly enhance,"?

**Response:** Yes, we have added the analysis of pressure gradient to explain the changes in winds (Figure 8, a new figure). The further explanation is as follows: "*Wind patterns show that easterly winds over the basin enhance while westerly wind over the plateau weaken (Figure S6 and Figure 7). We further compare the difference in the surface pressure between the baseline and sensitivity simulations, and find out that surface pressure over the plateau and the basin all decreases when the plateau warms by 2°C (Figure 8a and 8b). Over the plateau, the pressure drop has a decrease characteristic from west to east (Figure 8c), which results in a decreased pressure gradient and a weakened westerly wind. While in the basin, the pressure drop is less than the plateau. This leads to an increased pressure gradient from the basin to the plateau, inducing an intensified easterly wind. The enhanced easterly wind causes an increased transport of PM$_{2.5}$ from the basin to the plateau. On the other hand, the weakened*

*westerly wind and the enhanced easterly wind are convergent at the border between the plateau and the basin (Figure 7), jointly leading to an increase in PM$_{2.5}$ concentration at the eastern edge of the plateau. Additionally, northerly winds over the basin slightly enhance, conducive to diluting the air and reducing PM$_{2.5}$ concentration. Both easterly winds transport and northerly winds dilution are favorable for a reduction of PM$_{2.5}$ concentration in the basin.*" In lines 278 - 291.

[Figure]

**Figure 8** Comparison of spatial distributions of sea level pressure (SLP) between the (a) baseline simulation and (b) sensitivity simulation over the Tibetan Plateau and Sichuan Basin. (c) The SLPs over the plateau and basin decrease while the plateau becomes 2°C warming.

**Comment 5.** In line 293-295, similarly, could you further explain the mechanism of "a maximal temperature reduction located at 1.5 km to 3 km above the ground (Figure 9a)"?

**Response:** We have added the explanation: *"This is probably due to a sharp topography decrease (from ~ 5 km in the plateau to < 1 km in the basin) that leads to a warm plume via subsidence. In the basin, there is a decrease in the temperature from the surface to ~ 4 km, with a maximal temperature reduction (1 - 2°C) located at 1.5 km to 3 km above the ground (Figure 10a). We speculate that changes in the surface pressure can account for the maximal*

*temperature reduction here. After the 2°C warming, surface pressure decreases in the basin (Figure 8), which produces more convergent airflow (as shown in Figure 7). The strengthened convergent airflow induces an intensified ascending motion, conducive to a reduction of temperature in the basin. As a result, the zone where the maximal temperature drop appears, overlaps with the zone with the maximal ascending motion. Furthermore, the intensified updraft increases the vertical temperature gradient and the instability in the lower troposphere of the basin, thereby causing a higher PBL height than that in the non-warming case (Figure 10b). On the contrary, the change in vertical temperature profile leads to a decreased vertical temperature gradient and increased thermal stability in the lower troposphere of the plateau, in which the PBL height decreases.*

*On the other hand, the convergent airflows by a weakened westerly wind over the plateau and a strengthened easterly wind in the basin (shown in Figure 8) triggers an ascending motion on the east side of the plateau, which is also beneficial to the development of the PBL height in the basin. Consequently, the elevated PBL facilitates vertical diffusion, leading to a reduction in $PM_{2.5}$ concentration over the basin."* In Lines 313 - 332.

**Comment 6.** Related to comments 4 and 5, the paragraph from line 302-311 did not make very clear discussion on the changes of wind and temperature gradient. I suggest the com- parison of the changes of pressure-difference between TP and basin, and see the circulation changes could easily explain the issues in comments 4 and 5.

**Response:** Thanks for your suggestions, in the revised manuscript, we have re-written the paragraph, and the text is referring to the response to Comment 5.

**Comment 7.** I don't think the ascending motion in this study is similar to the plateau "heat pump" effect raised by Lau (2016)

**Response:** Yes, they are not the same. Here, we consider of the reviewer's comment, and have deleted this statement that the ascending motion in this study is similar to that in the EHP mechanism.

Elevated Heat Pump (EHP) hypothesis proposed by Lau and Kim (2006) illustrate that absorbing aerosols (dust and black carbon) heat up the air over the south slope of the Tibetan Plateau, inducing an ascending motion in lower troposphere and a positive temperature

anomaly in the mid-to-upper troposphere over the TP. According to the mass continuity principle, the air divergent in upper level and the air convergent in lower level, which further strengthens the upward motions. Under the circumstance, low-level convergence draws more warm and moist air from South Asia to increase monsoon rainfall. This thermodynamic mechanism shows that the heated plateau acts as an elevated heat pump.

In the present study, the temperature over the Tibetan Plateau rises by 2°C, which triggers an upward airflow on the eastern edge of the plateau. We would like to think that the role of the 2°C warming over the TP is similar to the positive temperature anomaly induced by absorbing aerosols in the EHP mechanism. Consequently, the 2°C warming leads to a convergent airflow and an ascending motion on the east edge of the plateau. The difference is that the EHP mechanism happens in the north-south direction, and our study explains the similar thermodynamic processes in the east-west direction.

**Technical corrections:**

**Comment 8.** I am misleading by the figure 6 in the first look and regards they are pie charts in percentage of species. Plot them as columns could be better.

**Response:** We have modified Figure 6 by a column chart, seen Figure 6 in the revised version.

[Figure]

**Figure 6** Comparison of chemical composition of $PM_{2.5}$ concentration between the baseline simulation (red bar) and sensitivity simulation (blue bar) over the Sichuan Basin.

**Comment 9.** Setting figure 11 as figure 10c is reasonable.

**Response:** We have combined Figure 10 and Figure 11 together, and labeled Figure 11 in the revised version.

[Figure]

**Figure 11** Comparison of spatial distributions of relative humidity (RH) between the (a) baseline simulation and (b) sensitivity simulation over the Tibetan Plateau and Sichuan Basin. (c) Spatial changes in RH after the plateau becomes 2°C warming, and the positive shows the RH increases while the negative shows the RH decreases.

---

## Author Response (AR3)

Oct 27, 2020

Dear Editor,

According to you and the reviewer's suggestion for technical corrections of our manuscript, we added some text to explain the model domain settings *in Section Model configuration and experiments*, and also discussed the model domain might be small while explaining the impact of atmospheric circulation related to the Tibetan Plateau on air quality **in Conclusions**. The details refer to the following response.

Thank you very much for handling our paper.

Best regards,
Xuexi Tie

**Reply to Anonymous Referee**

Thanks for the reviewer's helpful suggestions. We have given technical corrections to the size of the model domain in the revised manuscript.

**Technical Corrections**

The authors conclude that the warming TP or cooling TP will have similar effects on air quality in Sichuan Basin. However, I think that the simulation domain may be too small to reveal the effect of TP on air quality in Sichuan Basin. The Sichuan Basin is located nearby the boundary of the simulation domain. It is better to simulate the experiments at a wider range and make the Sichuan Basin and TP located in the center of domain. In this way, we can comprehensively see changes in atmospheric circulation over the TP and its effects on local circulation in Sichuan Basin.

Corrections: We have added the text to explain the model domain settings in Lines 140-146: "*Ideally, this study should set the Tibetan Plateau and the Sichuan Basin as the center of the model domain. However, considering the domain is too large, beyond the capability of our computer, we have to reduce the model domain. Nonetheless, to better simulate the atmospheric circulation over the plateau and its impact on air quality in the Sichuan Basin, we set the central location of the model domain at 95.0°E, 32.2°N over the plateau, and the simulation domain covers the Tibetan Plateau and the Sichuan Basin (Figure 1).*"

We also discussed the limitation of the model domain in Lines 476-484:"*Here we need to clarify that our model domain may be a little small to comprehensively atmospheric circulation pattern related to the Tibetan Plateau and the subsequent impact on air quality in the Sichuan Basin, though the domain covers the plateau and the basin. We notice that the basin is nearby the eastern boundary of the domain, in which local circulation might be influenced by the lateral condition. Therefore, as the plateau is likely to continue*

*warming, in-depth understanding to climate change on the Tibetan Plateau and long-term $PM_{2.5}$ measurements are required to validate the impact of the warming plateau on air quality in a larger spatial scale.*"

**The Warming Tibetan Plateau improves winter air quality in the Sichuan Basin, China**

Shuyu Zhao[1], Tian Feng[2], Xuexi Tie[1,3*], Zebin Wang[4]

[1]Key Laboratory of Aerosol Chemistry and Physics, SKLLQG, Institute of Earth Environment, Chinese Academy of Sciences, Xi'an, 710061, China

[2]Department of Geography & Spatial Information Techniques, Ningbo University, Ningbo, 315211, China

[3]Center for Excellence in Urban Atmospheric Environment, Institute of Urban Environment, Chinese Academy of Sciences, Xiamen, 361021, China

[4]Northwest Air Traffic Management Bureau, Civil Aviation Administration of China, Xi'an, 712000, China

Corresponding author: tiexx@ieecas.cn

**Key points**

The Tibetan Plateau is rapidly warming, and the temperature has risen by 2 ° C from

2013 to 2017.

A warming plateau leads to an enhanced easterly wind, an increased PBLH and a decreased RH in the Sichuan Basin.

The 2 ° C warming significantly reduces $PM_{2.5}$ concentration in the basin by 25.1 μg

$m^{-3}$, of which secondary aerosol is 19.7 μg $m^{-3}$.

**Abstract**

Impacts of global climate change on the occurrence and development of air pollution have attracted more attentions. This study investigates impacts of the warming Tibetan

Plateau on air quality in the Sichuan Basin. Meteorological observations and ERA- interim reanalysis data reveal that the plateau has been rapidly warming during the last

40 years (1979-2017), particularly in winter when the warming rate is approximately twice as much as the annual warming rate. Since 2013, the winter temperature over the plateau has even risen by 2 ° C. Here we use the WRF-CHEM model to lay emphasis on the impact of the 2 ° C warming on air quality in the basin. The model results show that the 2 ° C warming causes an enhanced easterly wind, an increase in the Planetary

Boundary Layer height (PBLH) and a decrease in the relative humidity (RH) in the basin. Enhanced easterly wind increases $PM_{2.5}$ transport from the basin to the plateau.

The elevated PBLH strengthens vertical diffusion of $PM_{2.5}$, while the decreased RH

significantly reduces secondary aerosol formation. Overall, $PM_{2.5}$ concentration is reduced by 17.5% (~25.1 μg m$^{-3}$), of which the reduction in primary and secondary aerosols is 5.4 μg m$^{-3}$ and 19.7 μg m$^{-3}$, respectively. These results reveal that the recent warming plateau has improved air quality in the basin, to some certain extent, mitigating the air pollution therein. Nevertheless, climate system is particularly complicated, and more studies are needed to demonstrate the impact of climate change on air quality in the downstream regions as the plateau is likely to continue warming.

**Keywords:** climate change, air quality, Tibetan Plateau, WRF-CHEM model

**1 Introduction**

The Tibetan Plateau is known as the third pole because of its high altitude and large area. It is also regarded as an important response region to the Northern Hemisphere, and even global climate due to its sensitivity to climate change. Previous studies on the Tibetan Plateau show that the region was experiencing warming in the second half of the 20th century, especially in the winter months (Kuang and Jiao, 2016; Liu and Chen, 2000; Rangwala et al., 2009). The warming plateau not only plays a significant role in driving the weather and climate change, as well as the ecological system, but also has an important impact on air quality in the downstream regions. Xu et al. (2016) suggest that the thermal anomaly over the Tibetan Plateau obviously increases haze frequency and surface aerosol concentration in central-eastern China.

However, the impacts of climate change on air quality in China are still unclear. Some researches hold the opinion that climate change induced by greenhouse gas emission increases severe haze occurrence and intensity in winter at Beijing, and its impact will continue in the future (Cai et al., 2017; Zou et al., 2017). Similarly, Xu et al. (2017) suggest that climate warming anomaly in the lower and middle troposphere over the continent around the Yangtze River Delta leads to more haze days in winter during recent decades. On the contrary, another opinion suggests that climate change in the past two decades is favorable for air pollution dispersion in northern China via enhancing mid-latitude cold surges in winter (Zhao et al., 2018). If cold surge is strong enough, pollutants would be transported to the downstream regions, causing a better air quality in the upstream region but a worse one in the downstream region. Thus, there may be regional differences in the impact of climate change on air quality.

Previous studies on air pollution in China are concentrated in the developed regions, such as the North China Plain, the Yangtze River Delta and the Pearl River Delta. Few studies have paid attention to the Sichuan Basin, although the region is undergoing severe air pollution, and mean $PM_{2.5}$ concentration is more than 110 μg m$^{-3}$ in winter (Qiao et al., 2019; Tao et al., 2017; Wang et al., 2018; Yang et al., 2011). Thus, it is necessary to explore the underlying causes that leads to air pollution in the Sichuan

Basin.

The Sichuan Basin locates in the downstream region of the Tibetan Plateau, and its weather conditions are obviously affected by the plateau (Duan et al., 2012; Hua, 2017;

Zhao et al., 2019). For instance, the foggy weather, southwest vortex and low-level shear line over the basin are closely associated with the plateau (Zhu et al., 2000). These changes in weather conditions induced by the plateau undoubtedly affect the development and dispersion of air pollution in the basin, because the huge terrain can trigger a thermodynamic forcing, which is of great importance for weather conditions in the surrounding regions (Bei et al., 2016; 2017; Zhao et al., 2015).

This study therefore focuses on how climate change on the Tibetan Plateau affects air quality in the Sichuan Basin in recent years. Section 3 analyzes the temperature change on the plateau in the past four decades, and especially emphasizes the change in recent five years. In Section 4, we design three sets of numerical simulations to calculate the impact of temperature change on air quality. One group includes two baseline simulations in two periods (January 2014 and January 2018), which are constrained by observed surface meteorological parameters and pollutant concentrations. The second group includes three sensitivity simulations during the 2013-2014 winter, which uses the same emission inventory and meteorological fields as the baseline simulation in

January 2014 except for a changed air temperature. We also set the third sensitivity simulation, in which the plateau is also warming, but on the basis of the period for the

2017-2018 winter. We compare the difference in $PM_{2.5}$ concentrations in these cases, and also calculate differences in meteorological parameters that include winds (wind speed and direction), air temperature, and relative humidity (RH), as well as the

Planetary Boundary Layer height (PBLH). Based on the differences in $PM_{2.5}$

concentration and meteorological parameters above, we finally explain the cause-to- effect relationship between a warming plateau and changes in the winds, PBLH and RH

in the Sichuan Basin. Moreover, we calculate the effect of the relationship on air quality in the basin.

**2 Data and Methods**

**2.1 Observations**

To ensure a robust result, we use two datasets of surface air temperature in this study.

One is the European Center for Medium-Range Weather Forecasts (ECMWF) ERA-

Interim monthly mean reanalysis data (1979-2018), obtained from the website of http://apps.ecmwf.int/datasets/, with the finest horizontal resolution of 0.125°×0.125°.

The other is hourly and monthly mean weather-station observations from the National

Oceanic and Atmospheric Administration (NOAA), available from http://gis.ncdc.noaa.gov/map/viewer/#app=clim&cfg=cdo&theme=hourly&layers=1

&node=gis.

Figure 1 shows the distribution of weather stations over the Tibetan Plateau, and these weather stations widely cover the entire plateau. Trends of annual mean and winter surface air temperature over the plateau are analyzed, and the winter is averaged over

3-month periods (December-January-February). Additionally, we use ambient air quality data to validate the model performance. Since 2013, the data are released by

Ministry of Environmental Protection, China at http://www.aqistudy.cn/, including hourly $PM_{2.5}$, CO, and $O_3$ mass concentrations. The monitoring stations for air quality are also shown in Figure 1.

**2.2 Model configuration and experiments**

A state-of-the-art regional dynamical and chemical model (WRF-CHEM model) is used in the study. Ideally, this study should set the Tibetan Plateau and the Sichuan Basin as the center of the model domain. However, considering the domain is too large, beyond the capability of our computer, we have to reduce the model domain. Nonetheless, to better simulate the atmospheric circulation over the plateau and its impact on air quality in the Sichuan Basin, we set the central location of the model domain at 95.0°E, 32.2°N

over the plateau, and the simulation domain covers the Tibetan Plateau and the Sichuan

[revised manuscript text omitted]

**4.2 Change in winter PM$_{2.5}$ concentration over the basin**

To examine impacts of a warming plateau on PM$_{2.5}$ concentration in winter in the basin, we set three levels of temperature increase of 0.5°C, 1.0°C and 2.0°C over the plateau.

Time series of PM$_{2.5}$ concentrations in these simulations (with and without the warming over the plateau) are respectively calculated. The results show that PM$_{2.5}$

concentration in the basin is significantly reduced (Figure 5). In comparison with three levels of temperature increase, the maximal reduction occurs in the case of 2°C

warming, with an average of 25.1 µg m$^{-3}$ ($p < 0.001$). Under the circumstance of the

2°C warming, the maximal hourly reduction reaches to 84.6 µg m$^{-3}$ (Figure S6a) and the maximal percentage reduction is about 64.4% (Figure S6b). We also calculate the changes in PM$_{2.5}$ concentration and its percentage under the influence of the 2°C

warming on the basis of January 2018 (Figure S7), of which the result is consistent with Figure S6, though there must inevitably be some differences in the magnitude.

Interestingly, the maximal reduction always occurs while PM$_{2.5}$ concentration reaches a peak value, which suggests that the impact of the warming plateau is extremely significant during the period of high $PM_{2.5}$ concentration. This result is similar to previous studies which also point out that extreme weather plays important roles in affecting air quality (De Sario et al., 2013; Hong et al., 2019; Tsangari et al., 2016;

Zhang et al., 2016). That is to say, the impact of the warming plateau on air quality is apt to be amplified in extremely high $PM_{2.5}$ concentrations.

To better understand the impact of a warming plateau on $PM_{2.5}$ concentration, we also calculate changes in $PM_{2.5}$ chemical composition in the basin. Both primary and secondary aerosols in $PM_{2.5}$ decreases more significantly with an increase in temperature increment (Figure 6), except for the nitrate due to its competition for ammonia with sulfate (Feng et al., 2018). As a result, the more sulfate is reduced under the case of the 2°C warming, the less nitrate is reduced. As shown in Figure 6, the warmer the plateau is, the more $PM_{2.5}$ concentration and its chemical composition in the basin decrease, suggesting that a warming plateau has increasing implications for air quality in the basin. Here we show that the maximal impact of the plateau under the case of the 2°C warming, in which secondary aerosol reduces by 19.7 μg m$^{-3}$, accounting for 78.5% of the total reduction, greatly larger than the reduction of primary aerosol. For example, the largest reduction is SOA, reducing from 23.2 μg m$^{-3}$ in the base case to 10.8 μg m$^{-3}$ in the 2°C warming case. The second reduction is sulfate (from 31.8 μg m$^{-3}$ to 28.6 μg m$^{-3}$). The next are nitrate and ammonium (22.3 μg m$^{-3}$

and 19.1 μg m$^{-3}$ in the base case, and 20.2 μg m$^{-3}$ and 17.5 μg m$^{-3}$ in the 2°C warming case). Significance testing of the difference in every chemical composition between the baseline simulation and the 2°C warming case is also given in Table S2. The *p*- values of most chemical composition in $PM_{2.5}$ are far less than 0.001 except that the

*p*-value of EC is 0.0011 (Table S2), implying for an extremely significant reduction of every chemical composition in $PM_{2.5}$ within the basin when the plateau warms by 2°C.

Thus, we emphasize the impact of the 2°C warming over the plateau on $PM_{2.5}$

concentration in the basin in our study. Meanwhile, we analyze the case for the 2017-

2018 winter, in which a similar change in $PM_{2.5}$ chemical composition is obtained when the plateau becomes 2°C warmer (Figure S8).

There are also significant changes in the spatial distribution of $PM_{2.5}$ concentration.

Figure 7 shows the spatial distribution of changes in surface $PM_{2.5}$ concentration and winds after 2°C warming over the plateau. Apparently, there is a larger decrease in

$PM_{2.5}$ concentration in the whole basin, and the maximal reduction is more than 30 µg

$m^{-3}$. By contrast, $PM_{2.5}$ concentration increases by 5 - 15 µg $m^{-3}$ at the eastern edge of the plateau. Wind patterns show that easterly winds over the basin enhance while westerly wind over the plateau weaken (Figure S5 and Figure 7). Enhanced easterly winds and weakened westerly winds are both in favor of the east-to-west transport of pollutants from the basin to the plateau. We also show changes in $PM_{2.5}$ concentration and winds under the cases of 0.5°C and 1.0°C warming in January 2014, consistent with the result of the 2°C warming, except that the reduction of $PM_{2.5}$ concentration and the change in wind speed are fewer (Figure S9a and Figure S9e). The case in

January 2018 (Figure S10a) is also similar to the result of the 2°C warming.

We further compare the difference in the surface pressure between the baseline and sensitivity simulations, and find out that surface pressure over the plateau and the basin all decreases when the plateau warms (Figure 8a and 8b). Over the plateau, the pressure drop has a decrease characteristic from west to east (Figure 8c, Figure S9b and Figure

S9f, Figure S10b), which results in a decreased pressure gradient and a weakened westerly wind. While in the basin, the pressure drop is less than the plateau. This leads to an increased pressure gradient from the basin to the plateau, inducing an intensified easterly wind. The enhanced easterly wind causes an increased transport of $PM_{2.5}$ from the basin to the plateau. On the other hand, the weakened westerly wind and the enhanced easterly wind are convergent at the border between the plateau and the basin (Figure 7, Figure S9a and Figure S9e, Figure S10a), jointly leading to an increase in

$PM_{2.5}$ concentration at the eastern edge of the plateau. Additionally, northerly winds over the basin slightly enhance, conducive to diluting the air and reducing $PM_{2.5}$

concentration. Both easterly winds transport and northerly winds dilution are favorable for a reduction of $PM_{2.5}$ concentration in the basin. In addition to the wind effect, there are also other important factors to produce the $PM_{2.5}$ reduction in the basin, such as the

PBLH and RH, which will be analyzed as follows.

**4.3 Impact of PBLH on $PM_{2.5}$ concentration**

Previous studies show that the PBL development plays an important role in diffusing pollutants (Miao et al., 2017; Su et al., 2018; Tie et al., 2015). Here we calculate the change in the PBLH due to the 2°C warming over the plateau, and then analyze the effect of the change in PBLH on $PM_{2.5}$ concentration in the basin.

Our results suggest that the warming plateau plays different roles in the PBL

development over the plateau and the basin. Due to the warming, the PBLH decreases in most areas of the plateau, but it increases over the basin (Figure 9, Figure S9c and

Figure S9g, Figure S10c). The maximal rise occurs under the case of the 2°C warming, by 50 - 200 m over the basin (Figure 9 and Figure S10c). As known, a shallow PBL

constrains $PM_{2.5}$ near the surface via suppressing vertical dispersion (Fan et al., 2011;

Iversen, 1984). Conversely, a deep PBL is favorable for $PM_{2.5}$ diffusion. Thus, we explore the underlying cause that leads to the difference in the PBLH in the domain.

The PBLH is strongly related to the changes in vertical temperature and wind, Figure

10 and Figures S11-12 display vertical profiles of changes in temperature and winds in the plateau and the basin. Results show that the warming causes a maximal warm layer around 1 km above the ground of the plateau. Noticeably, the warm layer acts as a dome covering 4.5 km above the Sichuan Basin (Figure 10a, Figure S11a and Figure S11c,

Figure S12a). Xu et al. (2017) also finds out a significant warm plume extending from the plateau to the downstream Sichuan Basin and Yangtze River Delta by use of

NCEP/NCAR reanalysis data. We suggest that this is probably due to a sharp topography decrease (from ~ 5 km in the plateau to < 1 km in the basin) which leads to a warm plume via subsidence. In the basin, there is a decrease in the temperature from the lower troposphere to ~ 4 km, with a maximal temperature reduction (0.5 - 2°C)

located at 1.5 km to 3 km above the ground (Figure 10a, Figure S11a and Figure S11c,

Figure S12a). We speculate that changes in the surface pressure can account for the maximal temperature reduction here. After the warming, surface pressure decreases in the basin (Figure 8, Figure S9b and Figure S9f, Figure S10b), which produces more convergent airflow (as shown in Figure 7, Figure S9a and Figure S9e, Figure S10a).

The strengthened convergent airflow induces an intensified ascending motion, conducive to a reduction of temperature in the basin. As a result, the zone where the maximal temperature drop appears, overlaps with the zone with the maximal ascending motion. Furthermore, the intensified updraft increases the vertical temperature gradient and the instability in the lower troposphere of the basin, thereby causing a higher PBLH

than that in the non-warming case (Figure 10b, Figure S11b and Figure S11d, Figure

S12b). On the contrary, the change in vertical temperature profile leads to a decreased vertical temperature gradient and increased thermal stability in the lower troposphere of the plateau, in which the PBLH decreases.

On the other hand, the convergent airflows by a weakened westerly wind over the plateau and a strengthened easterly wind in the basin triggers an ascending motion on the east side of the plateau (Figure 10a, Figure S11a and Figure S11c, Figure S12a), which is also beneficial to the development of the PBLH in the basin. Consequently, the elevated PBL facilitates vertical diffusion, leading to a reduction in $PM_{2.5}$

concentration over the basin.

**4.4 Effect of RH on $PM_{2.5}$ concentration**

In addition to the PBLH, ambient RH is a key factor for secondary aerosol formation (Tie et al., 2017; Wang et al., 2016). Previous studies indicate that aerosol hygroscopic growth cannot occurs until the humidity exceeds 50% (Liu et al., 2008). When the humidity is greater than 60%, hygroscopic growth factor of urban aerosol increases significantly with humidity (Liu et al., 2008).

We examine the influence of in the RH change induced by a warming plateau on $PM_{2.5}$ concentration in the basin. Results show that there is remarkable change in RH in the basin due to the warming of the plateau (Figure 11, Figure S9d and Figure S9h, Figure S10d). In the baseline simulation, the RH varies in the range of 40% - 80% over the basin (Figure 11a). However, the RH varies from 50% to 70% in the 2 °C warming simulation (Figure 11b), suggesting that the basin becomes drier when the plateau is warmer.

The RH comparison between these numerical simulations reveals that the warming causes a decrease in the RH within the basin (Figure 11c, Figure S9d and Figure S9h, Figure S10d). These changes in RH have critical effects on the secondary aerosol formation. As explained by Tie et al. (2017), the reduction of RH (especially during the stage of RH from 80% to 70%) causes a significant decrease of hygroscopic growth on the aerosol surface, resulting in less water surface for producing secondary aerosol, such as sulfate and nitrate. As a result, $PM_{2.5}$ concentration decreases in the basin. There are also some fingerprints of the RH's effect on $PM_{2.5}$ concentration. Firstly, the spatial distributions of RH reduction and $PM_{2.5}$ concentration reduction have similar patterns (Figure 11c and Figure 7, Figure S9a and Figure S9d, Figure S9e and Figure S9h, Figure S10a and Figure S10d), and the region with more humidity decrease overlaps the region with more $PM_{2.5}$ decreases. Secondly, as shown in Figure 6, the changes in $PM_{2.5}$ compositions indicate that the reduced $PM_{2.5}$ concentration is mainly caused by the decrease in secondary aerosol concentration. Therefore, the RH change also plays an important role for $PM_{2.5}$ concentration in the basin.

**5 Conclusions**

ERA-interim reanalysis data and observation records at 10 weather stations show that the Tibetan Plateau is significantly warming during the past four decades (1979-2017), particularly in winter. The temperature increase rate is 0.5°C decade$^{-1}$ to 1.0°C decade$^{-1}$

$^{1}$ in winter, approximately twice as much as the increase rate of annual mean temperature. In recent 5 years (2013-2017), the central plateau is significantly warming with an increase rate of 1.0°C yr$^{-1}$, encompassing the warming rate during the entire 40

years. Rapid warming has caused the winter temperature to increase by an average of

2°C over the entire plateau from 2013 to 2017.

The WRF-Chem model is used to assess the impact of a warming plateau on air quality over the downstream Sichuan Basin. The most significant impact of the plateau on

$PM_{2.5}$ concentration in the basin occurs under the case of the 2°C warming. Through an enhanced horizontal transport, a reduced RH and an increased PBLH, the warming plateau significantly reduces $PM_{2.5}$ concentration in the basin. A larger pressure gradient from the basin to the plateau is favorable for an east-to-west transport for pollutants within the basin. A lower ambient RH decreases aerosol hygroscopic growth, which weakens secondary aerosol formation and leads to a significant reduction in secondary aerosol concentration. Moreover, the warming induces an increase in vertical temperature gradient over the basin, strengthening turbulence mixing and elevating

PBLH. The elevated PBLH is favorable for vertical diffusion that causes a reduction of

$PM_{2.5}$ in the basin. Additionally, the uplift effect by an enhanced ascending motion at the eastern edge of the plateau also contributes to $PM_{2.5}$ reduction within the basin.

In summary, the warming over the plateau in recent five years comprehensively induces a rising PBLH and a drying ambient air over the basin, which greatly reduces $PM_{2.5}$

secondary compositions. On average, $PM_{2.5}$ concentration reduces by 25.1 μg m$^{-3}$ on the basis of the 2013-2014 winter, of which the primary and secondary aerosols decrease by 5.4 μg m$^{-3}$ and 19.7 μg m$^{-3}$, respectively. Here we need to clarify that our model domain may be a little small to comprehensively atmospheric circulation pattern related to the Tibetan Plateau and the subsequent impact on air quality in the basin, though the domain covers almost the whole plateau and the basin. We notice that the basin is nearby the eastern boundary of the domain, in which local circulation might be influenced by the lateral condition. Therefore, as the plateau is likely to continue warming, in-depth understanding to climate change on the Tibetan Plateau and long- term $PM_{2.5}$ measurements are required to validate the impact of the warming plateau on air quality in a larger spatial scale.

[revised manuscript text omitted]